

# Gibberellic acid-mediated transcriptional divergence underlies cold stress adaptation in two diploid cotton species

Dong Wang[1,*], Juyun Zheng[2,*], Ke Liu[3], Yanchao Xu[4] and Dingsha Jin[5]

[1] Xinjiang Jinfengyuan Seed Industry Co., LTD, Urumuqi, China
[2] Cotton Research Institute of Xinjiang Uygur Autonomous Region Academy of Agricultural Sciences, Xinjiang Key Laboratory of Cotton Genetic Improvement and Intelligent Production, Urumuqi, China
[3] Agricultural Development Service Center of Shihezi, Shihezi, China
[4] National Nanfan Research Institute (Sanya), Chinese Academy of Agricultural Sciences, Sanya, China
[5] Hainan Academy of Agricultural Sciences, Sanya Research Institute, Sanya, China
* These authors contributed equally to this work.

Corresponding authors
Yanchao Xu,
xuyanchao2016@163.com
Dingsha Jin, jindingsha@163.com

## ABSTRACT

**Background:** The diploid cotton species *Gossypium thurberi* (D1) and *Gossypium trilobum* (D8) exhibit significant divergence in cold stress tolerance despite their close phylogenetic relationship.

**Methods:** To explore the genetic basis of this difference, we conducted a comparative transcriptomic analysis under cold stress at 4 °C, identifying 697 and 311 species-specific differentially expressed genes (DEGs) in *G. thurberi* and *G. trilobum*, respectively. Functional enrichment analysis was performed to investigate the biological pathways associated with these DEGs. Additionally, hormone levels, particularly gibberellic acid (GA), were measured to assess their role in cold stress responses.

**Results:** The DEGs in both species were significantly enriched in the "hormone signal transduction" pathway, highlighting the importance of hormonal regulation in cold adaptation. Distinct trends in GA levels were observed between *G. thurberi* and *G. trilobum*, with GA strongly correlated with species-specific DEGs. *G. thurberi* demonstrated greater cold tolerance than *G. trilobum*, likely due to a more robust GA-regulated response.

**Conclusion:** These findings indicate that expression divergence in GA-mediated pathways between sister species has driven adaptive evolution in cold stress tolerance. This study not only advances our understanding of cold adaptation mechanisms in cotton but also provides genetic insights for improving cold tolerance in cultivated varieties through targeted breeding and genetic engineering.

# INTRODUCTION

Cold stress significantly impacts crop growth, yield, and quality (*Chen et al., 2015*; *Si et al., 2018*). Cotton (*Gossypium* spp.), a globally important fiber crop and a key source of edible

oil and protein, originates from tropical and subtropical regions, making it particularly sensitive to low temperatures throughout its growth cycle, especially during germination and seedling stages (*Ren et al., 2018*; *Stewart & Guinn, 1971*). Chilling stress induces severe damage, including wilting, leaf discoloration, premature senescence, incomplete maturity, and even plant death (*Rikin, 1992*). The genus *Gossypium* comprises four cultivated species (*G. arboreum, G. herbaceum, G. hirsutum,* and *G. barbadense*) and 47 wild species (*Birchler, 2012*). Upland cotton (*G. hirsutum L.*) accounts for approximately 90% of global production (*Zhang et al., 2015*). However, intensive breeding for yield and fiber quality has led to reduced genetic diversity within cultivated cultivars (*Xu et al., 2019*). In contrast, wild *Gossypium* species, shaped by natural selection, exhibit rich genetic diversity and possess desirable traits such as superior fiber quality and resistance to abiotic and biotic stresses (*Ditta et al., 2018*). Plant responses to cold stress involve complex regulatory mechanisms at multiple levels (*Park et al., 2015*). Over the past decades, significant progress has been made in elucidating cold-responsive pathways at the transcriptional level (*Liao et al., 2019*; *Lu et al., 2017*; *Xu et al., 2018*). Among wild cotton species, *G. thurberi* (D genome), native to Arizona, USA, has adapted to its local climatic conditions and exhibits remarkable cold tolerance, surviving temperatures as low as −6 °C (*Cai et al., 2019*). In contrast, *G. trilobum* (also D genome) is highly susceptible to cold stress throughout its growth stages. Despite their close phylogenetic relationship-diverging approximately 0.7 million years ago with a mutation rate of $x \times 10^{-8}$ substitutions per site per year (*Grover et al., 2019*). These two species exhibit pronounced differences in cold adaptation. This contrast makesthem ideal models for investigating the genetic and molecular basis of cold tolerance in cotton.

Recent studies have highlighted the potential of wild cotton species as valuable reservoirs of genetic diversity and stress-resistant traits (*Cai et al., 2019*; *Chen et al., 2020*; *Wei et al., 2017*; *Xu et al., 2020*). To elucidate the genetic mechanisms underlying the contrasting cold tolerance of *G. thurberi* and *G. trilobum*, we conducted RNA sequencing (RNA-seq) on cold-treated leaves of both species. Comparative transcriptomic analyses were performed using the high-quality *G. raimondii* genome as a reference (*Paterson et al., 2012*), enabling the identification of interspecific orthologous genes and their expression patterns under cold stress. Our findings revealed significant enrichment of differentially expressed genes (DEGs) in the "plant hormone signaling pathway," with gibberellic acid (GA) playing a pivotal role in cold stress responses. Hormonal assays further confirmed species-specific differences in hormone levels, suggesting that divergence in hormonal regulation contributes to their contrasting cold tolerance. These results provide valuable insights into the genetic and molecular mechanisms of cold adaptation in cotton and offer a theoretical foundation for improving abiotic stress tolerance in cultivated cotton through the utilization of wild genetic resources.

# MATERIALS AND METHODS

## Plant material and cold stress conditions

Diploid wild cotton species *G. trilobum*, exhibiting varying sensitivities to low temperature were selected for this study. Over 200 cotton seeds with uniform and vigorous growth

potential were carefully selected. The seeds were surface sterilized using 8% sodium hypochlorite for 30–60 min followed by thorough rinsing with sterile water. Subsequently, germination was carried out in nutrient-rich soil at 26 °C under long-day conditions with a 16-h light/8-h dark cycle with a light intensity of ~16,000 LUX and 75% relative humidity. Once the seedlings had developed three true leaves and one heart-shaped leaf, 100 seedlings were exposed to low temperature stress (4 °C), while the remaining 100 seedlings were maintained at normal temperature (26 °C) as a control group under the same long-day conditions.

## Measurement of physiological and biochemical indices

To assess the physiological and biochemical responses of the cotton plants to cold stress, various parameters related to oxidants, antioxidant enzymes, and plant hormone levels were measured. Lipid peroxidation was quantified by measuring malondialdehyde (MDA) content usingthiobarbituric acid reactive substances (TBARS) assay (*Kinchen et al., 2018*). Superoxide dismutase (SOD) activity, a key antioxidant defense enzyme against low-temperature stress, was determined follwed the protocol described by *Katsyuba et al. (2018)*. Peroxidase (POD) activity was measured using the method of *Hu, Zhang & Yang (2019)*. The contents of proline (PRO), soluble protein (SP), and soluble sugar (SS) were measured following the methods described by *Eich et al. (2000)*, with the assistance of corresponding assay kits (BC0290; Beijing Solarbio Science & Technology Co., Ltd., Beijing, China). The contents of abscisic acid (ABA; CSB-E09159Pl), ethylene (ETH; MBS8806823), gibberellic acid (GA; EKU04370-96T), and indoleacetic acid (IAA; EKU04915-96T) in leaves were quantified using the enzyme-linked immunosorbent assay (ELISA) method. For each measurement, 0.5 grams of leaf samples were used, and the ELISA was performed using the Infinite M1000 (TECAN, Männedorf, Switzerland) (*Boucher et al., 2016*). Each sample was analyzed with three biological replicates.

## RNA extraction, cDNA library preparation and RNA-Seq

Total RNA was extracted from *G. thurberi* and *G. trilobum* samples using the cetyl trimethylammonium bromide (CTAB)-acidic phenol extraction method (*Jordon-Thaden et al., 2015*). The purity of the extracted RNA was evaluated using the NanoPhotometer spectrophotometer (IMPLEN, Westlake Village, CA, USA), and its concentration was measured using the Qubit RNA Assay Kit in a Qubit 2.0 Fluorimeter (Life Technologies, Carlsbad, CA, USA). The integrity of the RNA was assessed using the RNA Nano 6000 Assay Kit on the Agilent Bioanalyzer 2100 system (Agilent Technologies, Santa Clara, CA, USA). For each sample, 1 µg of high-quality RNA was used for cDNA library preparation. Sequencing libraries were generated using the NEBNext UltraTM RNA Library Prep Kit for Illumina (NEB, Ipswich, MA, USA) according to the manufacturer's instructions. Unique index codes were added to identify each sample. Library clustering index was performed on a cBot Cluster Generation System using the TruSeq PE Cluster Kit v4-cBot-HS (Illumina, San Diego, CA, USA), following the provided protocol. The clustered libraries were then sequenced on an Illumina Hiseq 2500 platform, generating paired-end reads. Each sample was analyzed with three biological replicates.

## Reads mapping, transcript assembly, and differential expression

The RNA-seq data of *G. thurberi* had been previously published (*Cai et al., 2019*). Raw reads obtained from RNA-Seq were processed to remove adapter sequences, poly-N sequences, and low-quality reads, resulting in high quality clean reads. The quality of the clean data was evaluated by calculating Q30, GC-content, and sequence duplication levels. These clean reads were then mapped to the reference genome sequence of *G. raimondii*. Only reads with a perfect match or a single mismatch were retained for further analysis and annotation. The Tophat2 software (*Kim et al., 2013*) was employed for read mapping, and the mapped reads were assembled using Cufflinks software (*Ghosh & Chan, 2016*). Gene expression levels were quantified as fragments per kilobase of transcript per million fragments mapped (FPKM) (*Shahriyari, 2019*). Cuff quant and Cuff form, components of the Cufflinks suite, were used to estimate gene expression levels were estimated based on the mapped reads with FPKM as the index. EBSeq software was used to identify differentially expressed genes (DEGs) with a fold change ≥ 2 and a false discovery rate (FDR) ≤ 0.01 (*Leng et al., 2013*). The FDR was corrected using the Benjamini–Hochberg method for *P*-values (α ≤ 0.05). The statistical power of this experimental design was calculated as 0.827 using RNASeq Power (*Hart et al., 2013*) is 0.827. The raw transcriptomic data are publicly available at the National Center for Biotechnology Information (NCBI) under the accession number PRJNA554553.

## Gene functional annotation, GO enrichment analysis, KEGG pathway enrichment analysis

Gene function annotation was conducted by aligning sequences to multiple databases, including Nr (NCBI non-redundant protein sequences), Nt (NCBI non-redundant nucleotide sequences), Pfam (Protein family), KOG/COG (Clusters of Orthologous Groups of proteins), Swiss-Prot (a manually annotated and reviewed protein sequence database), KO (KEGG Ortholog database), and GO (Gene Ontology). Gene Ontology enrichment analysis of the differentially expressed genes (DEGs) was performed using the GOseq R package (*Ghosh & Chan, 2016*), which adjusts for gene-length bias in DEGs based on the Wallenius non-central hyper-geometric distribution. The Kyoto Encyclopedia of Genes and Genomes (KEGG) (*Du et al., 2014*) was used to explore high-level biological functions and interactions at the cellular level, organism, and ecosystem levels, leveraging molecular data derived from genome sequencing and other high-throughput experimental technologies (http://www.genome.jp/kegg/). The KOBAS software (*Xie et al., 2011*) was utilized to test the statistical enrichment of differential expression genes in KEGG pathways.

## Validation of RNA-seq data by RT-qPCR

To validate the differential gene expression detected by Illumina RNA-Seq, real-time reverse transcription-polymerase chain reaction (RT-qPCR) was carried out. A total of 30 genes were randomly selected, including 10 up regulated, 10 downregulated, and 10 non expressed genes, based on their FPKM values at different time points. RT-qPCR was performed using a BioRad CFX96 Real-Time system with the LightCycler FastStart DNA

Master SYBR Green I kit (Roche, Basel, Switzerland). Each sample was analyzed in three biological replicates.

## Statistical analysis

Data processing was performed using Microsoft Excel. Bar graphs were generated using GraphPad Prism version 9.0 (GraphPad Software, San Diego, CA, USA).

# RESULTS

## Effects of cold stress on *G. thurberi* and *G. trilobum*

The cold tolerance of *G. thurberi* and *G. trilobum* was initially evaluated through a comprehensive analysis of physiological traits. After cold stress treatment, significant differences were observed in the activities of antioxidant enzymes and the contents of osmotic adjustment substances between the two cotton species. Three hours after cold stress, *G. trilobum* exhibited a relatively higher fold change in superoxide dismutase (SOD) activity. However, at twelve hours, *G. thurberi exhibited* significantly lower levels of peroxidase (POD) and proline (PRO) than *G. trilobum*. These results indicate that *G. thurberi* maintains a more balanced regulation of SOD, PRO, and POD, indicating stronger cold tolerance. Moreover, of the gradual increase in soluble sugar (SS) content in both species throughout the stress period highlights the crucial role of SS as a protective substance against cold stress (Fig. 1).

## RNA-seq analysis

The cDNA libraries of *G. thurberi* and *G. trilobum* were sequenced using the Illumina Hiseq 2500 platform, generating paired end reads. The raw sequencing data were initially filtered based on the base calling accuracy (Q30), resulting in 32.77–43.66 million high quality clean. These clean reads were then mapped to the sequenced genome of *G. raimondii* using the Tophat2 software. The mapping rate ranged from approximately 74.21% to 78.25%, and 22.91–30.83 million uniquely mapped reads were retrieved (Table 1). Based on these mapping results, 23,726 and 23,820 genes were identified in *G. thurberi* and *G. trilobum*, respectively. Functional annotation through sequence alignment with multiple databases, including NR, Swiss-Prot, GO, COG, KOG, Pfam, and KEGG, enabled effective annotation of more than 98% of the identified genes.

## Novel gene prediction

Novel genes that were not annotated in the reference genome but were expressed in the experimental materials were successfully predicted. Based on the mapping results, 482 and 470 novel genes were identified in *G. thurberi* and *G. trilobum*, respectively. Of these, 445 genes were successfully annotated through sequence alignment using databases such as GO, KEGG, Swissprot, and NR. Comparative analysis revealed that approximately half of these navel genes were also present in *G. arboreum*. Some new genes were detected in *Theobroma cacao*, and a small number were found in other species, including *Vitis vinifera*, *Citrus sinensis*, *Malus domestica*, *Eucalyptus grandis*, *Prunus persica*, *Oryza sativa*, *Populus trichocarpa*, and *Medicago truncatula* (Fig. 2).

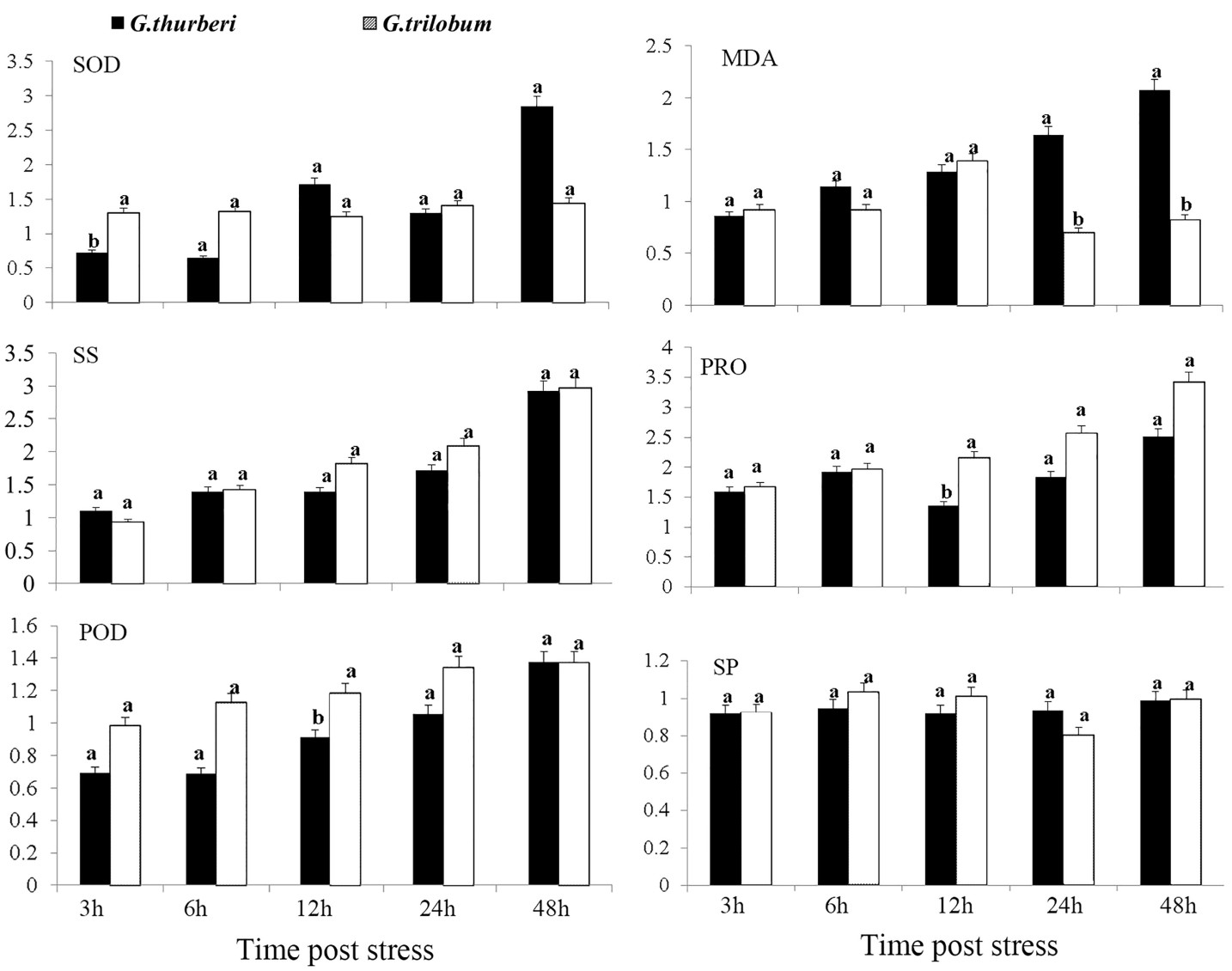

**Figure 1 Physiological analysis of *G. thurberi* and *G. trilobum* in response to various durations of cold stress.** (A–F) SOD, MD, POD, SS, PRO, and SP fold changes at different times post cold stress. *G. thurberi* and *G. trilobum* seedlings grown under normal conditions were used as controls. At least three biological replicates were used. Multiple comparisons were performed with significant difference in different letter at $P < 0.05$ level; Error bars represent SD.

## Expression divergence between the two sister species under cold stress

To further explore the relationship between cold adaptation and species differentiation, an in-depth analysis of the gene expression divergence between *G. thurberi* and *G. trilobum* was performed. Among the 23,128 mapped genes, approximately 58.8% exhibited similar expression levels across all growth stages in both species, reflecting their close phylogenetic relationship. However, differentially expressed genomic regions predominantly

**Table 1 Mapping result of RNA-seq reads.**

| Sample | Clean read (M) | Mapped read (M) | Mapped ratio (%) | %≥Q30 |
|---|---|---|---|---|
| ThCK0 | 43.66 | 34.17 | 78.25 | 91.59 |
| ThCK6 | 43.62 | 33.1 | 75.88 | 91.33 |
| ThCK12 | 34.48 | 26.04 | 75.53 | 92.16 |
| ThCS6 | 41.88 | 32.39 | 77.33 | 92.03 |
| ThCS12 | 34.98 | 26.9 | 76.91 | 92.02 |
| TrCK0 | 40.08 | 31.53 | 78.68 | 91.66 |
| TrCK6 | 32.77 | 24.77 | 75.59 | 92.31 |
| TrCK12 | 41.63 | 30.89 | 74.21 | 91.94 |
| TrCS6 | 42.42. | 32.76 | 77.23 | 91.55 |
| TrCS12 | 38.52 | 29.53 | 76.66 | 92.22 |

**Note:**
CK and CS, Control and cold stress treatment.

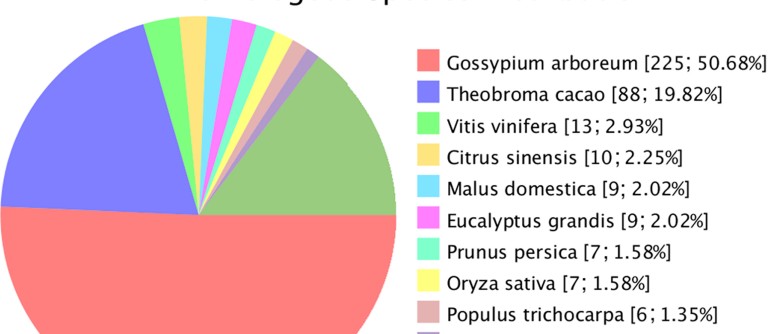

**Figure 2 Species of similar sequences with the new gene.** Different color represents different species.

concentrated at the terminal regions of the chromosomes. Genes that showed differential expression at all stages between the two species were defined as conserved differentially expressed genes (DEGs) (Fig. 3), while those that had no expression differences under normal conditions but exhibited expression changes under cold stress were designated as inducible DEGs. A total of 350 conserved DEGS were identified, showing significant enrichment in GO terms related to ADP binding (GO:0004353, *P*-value < 0.00025), hormone response (GO:0009725, *P*-value < 0.00028), and catalytic activity (GO:0003824, *P*-value < 0.00031). Additionally, 454 inducible DEGs were identified, which were mainly enriched in GO terms such as cellular response to phosphate starvation (GO:0016036, *P*-value < 8.0e−6) and plant-type cell wall organization (GO:0009664, *P*-value < 1.8e−5).
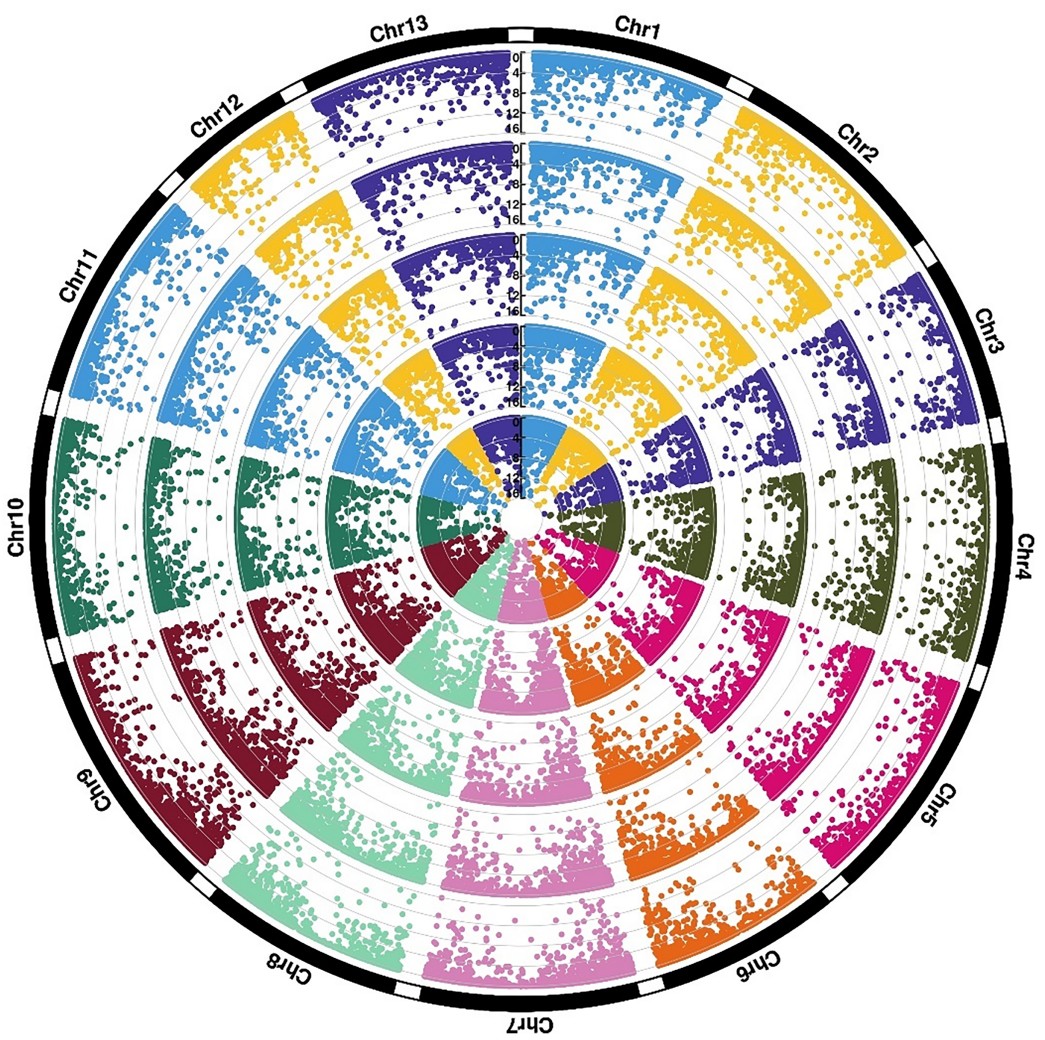

**Figure 3 Expression divergence of two sister species under cold stress.** Rings (from outer ring to inner ring) show −log10(FDR) values of CK0 (ThCK0 *vs.* TrCK0), CK6 (ThCK0 *vs.* TrCK0), CK12 (ThCK0 *vs.* TrCK0), CS6 (ThCK0 *vs.* TrCK0), CS12 (ThCK0 *vs.* TrCK0).

## Identification, annotation, and enrichment analysis of differentially expressed genes (DEGs)

In *G. thurberi*, a total of 4,227 DEGs were identified at least at one of the two time points (6 and 12 h) following cold stress. Specifically, 2,639 DEGs were detected at 6 h and 2,844 DEGs at 12 h, with 1,305 DEGs common to both time points (Fig. 4A). In *G. trilobum*, 4,174 DEGs were identified, with 2,139 and 2,740 DEGs at 6 and 12 h, respectively, and 697 common DEGs (Fig. 4B). A Venn diagram was used to summarize these transcripts, showing that the number of DEGs in *G. thurberi* and *G. trilobum* was similar, and most DEGs in both species were detected 12 h after cold stress, indicating the close genetic relationship between them.
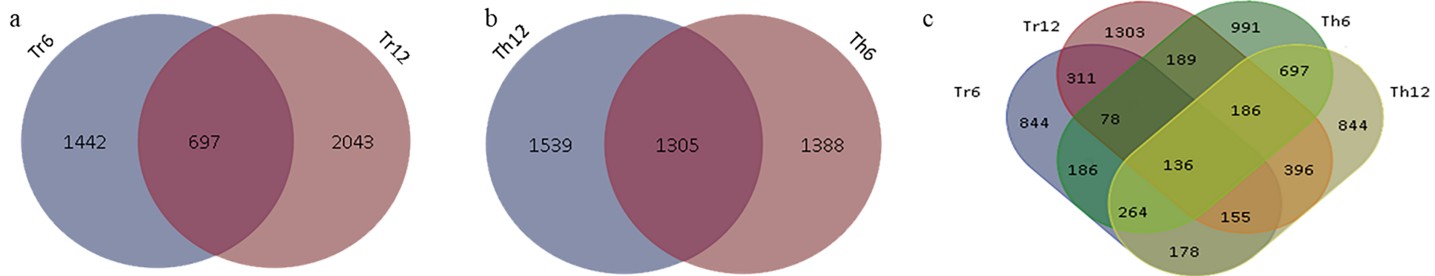

**Figure 4** **Summary of DEGs in *G. thurberi* and *G. trilobum*.** (A) Number of DEGs in *G. trilobum*. (B) Number of DEGs in *G. thurberi*. (C) Number of common DEGs between *G. thurberi* and *G. trilobum*.

| Table 2 DEGs annotation of *G. thurberi* and *G. trilobum* by different database. | | |
|---|---|---|
| Database | Annotated genes in of *G. trilobum* | Annotated genes in of *G. thurberi* |
| GO | 2,717 (64.97%) | 2,716 (64.18%) |
| KEGG | 926 (22.14%) | 805 (19.02%) |
| KOG | 2,230 (53.32%) | 2,169 (51.25%) |
| Swissprot | 3,419 (81.76%) | 3,427 (80.98%) |
| NR | 4,168 (99.67%) | 4,221 (99.74%) |
| NT | 4,126 (98.66%) | 4,175 (98.65%) |
| All | 4,174 (99.81%) | 4,227(99.88%) |

To identify key candidate genes involved in cold tolerance, 136 DEGs were found to be common between the two species at the two time points (Fig. 4C). The expression patterns of these 136 DEGs revealed that 51 genes were up-regulated, 67 genes were down-regulated, 17 genes had different expression regulation patterns in *G. thurberi* and *G. trilobum*, and one gene had different expression regulation patterns at 6 and 12 h. These 136 DEGs can serve as key genes for studying the differences in cold tolerance between *G. thurberi* and *G. trilobum*.

To better understand the biological functions of the DEGs, annotation and enrichment analyses were performed using databases such as Gene Ontology (GO), Eukaryotic Orthologous Groups of proteins (KOG), Kyoto Encyclopedia of Genes and Genomes (KEGG), NCBI non-redundant protein sequences (NR), NCBI non-redundant nucleotide sequences (NT), and Swiss-Prot. In this study, 4,232 and 4,182 DEGs from *G. thurberi* and *G. trilobum*, respectively, were used for analysis, with annotation rates of 99.88% and 99.81%, respectively (Table 2).

GO enrichment analysis was carried out using the GOseq R package based on the Wallenius non-central hyper-geometric distribution. In *G. thurberi* and *G. trilobum*, 2,716 (64.18%) and 2,717 (64.97%) DEGs were annotated, respectively. The results showed that most DEGs were enriched in biological processes such as "metabolic process", "cellular process", "single-organism process", and "response to stimulus", as well as in GO terms related to cellular components such as "cell", "cell part", "membrane", "organelle", and

molecular functions such as "catalytic activity" and "binding" (Figs. S1A, S1C). GO enrichment analysis also indicated that most genes responded to stresses such as cold, salt, water deprivation, and cadmium ion exposure.

KOG annotation revealed that, 2,230 (53.32%) and 2,169 (51.25%) DEGs in *G. thurberi* and *G. trilobum*, respectively, were enriched in categories such as "signal transduction mechanisms", "Posttranslational modification, protein turnover, chaperones", and "carbohydrate transport and metabolism" (Figs. S1B, S1D). KEGG pathway analysis showed that 805 (19.02%) and 926 (22.14%) DEGs in *G. thurberi* and *G. trilobum*, respectively, were mapped to known pathways, with the majority involved in the "Plant hormone signal transduction (ko04075)" pathway (Tables S2 and S3). This result is consistent with previous studies, highlighting the crucial role of plant hormones in stress responses. In addition, pathways such as "Starch and sucrose metabolism", "Glycolysis/ Gluconeogenesis", "Glutathione metabolism", "Purine metabolism", "Porphyrin and chlorophyll metabolism", and "Photosynthesis" also play important roles in regulating low-temperature stress.

GO enrichment analysis was also performed on the 136 common DEGs between the two species produced similar results, with most genes associated with "cell", "cell part", "membrane", "organelle catalytic activity", "binding", "metabolic process", "cellular process", "single-organism process", and "response to stimulus" (Fig. S2A). KOG enrichment analysis (Fig. S2B) showed similar trends in the enrichment results of all DEGs. Functional classification revealed their involvement in critical biological processes such as transcription factor regulation, signal transduction mechanisms, and carbohydrate transport and metabolism.

## DEGs involved in transcription factor regulation

Transcription factors (TF) are integral components of plants' regulatory networks involved in response to abiotic stresses (*Bai et al., 2015*; *Cai et al., 2019*). In this study, 23 out of 136 key DEGs exhibited TF activity.

After cold stress treatment (6 and 12 h), expression analysis in *G. thurberi* and *G. trilobum* revealed that 16 TFs were upregulated, while six were down-regulated (Fig. 5A). These TFs were classified into several well-known families, including MYB family (eight genes), WRKY family (three genes), bHLH family (three genes), and NAC family (two genes).

Previous studies have firmly established the significant roles of the MYB, WRKY, bHLH, and NAC families in enhancing plants' tolerance to abiotic stresses. For instance, the MYB family is known to regulate a wide range of stress responsive genes, while the WRKY family is often involved in the transcriptional regulation of defense related genes during stress responses. The bHLH family participates in various physiological processes, including stress adaptation, through binding to specific DNA sequences, and the NAC family is crucial for plant development and stress tolerance by modulating gene expression.

In addition, the AP2-EREBP family is a key TF family in plants' response to low-temperature stress. In both *G. thurberi* and *G. trilobum*, the *Gorai.002G251500* gene, belonging to the AP2-EREBP family, was up-regulated at 6 and 12 h after cold stress. This

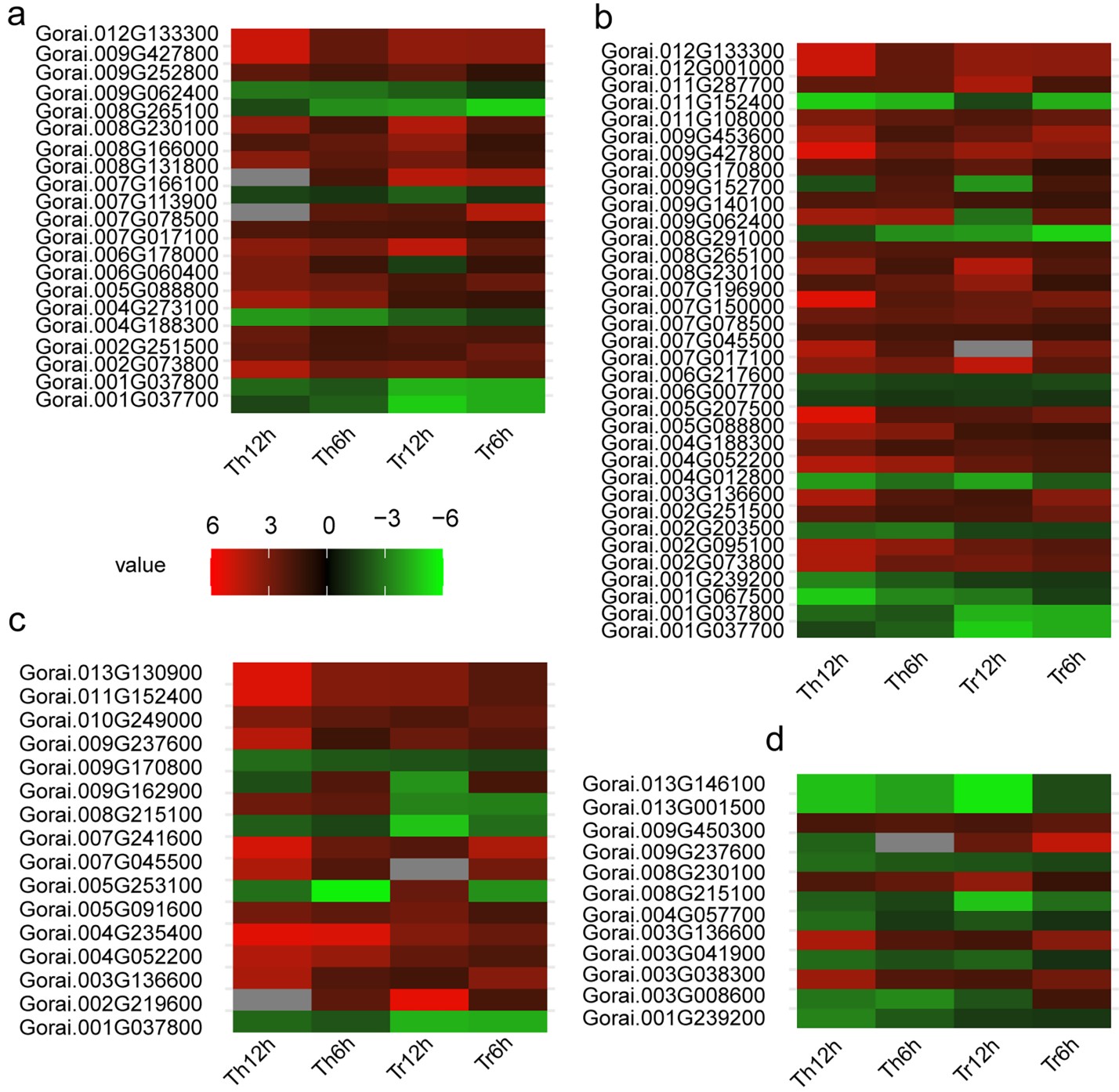

**Figure 5 Heat map of DEGs in *G. thurberi* and *G. trilobum*.** (A) Heat map depicting log2 (fold change) of transcription factor of 136 key DEGs; (B) heat map depicting log2 (fold change) of signal relative genes of 136 key DEGs; (C) heat map depicting log2 (fold change) of ROS relative genes of 136 key DEGs. (D) Heat map depicting log2 (fold change) of carbohydrate transport and metabolism relative genes of 136 key DEGs. Gs. Red and green colors indicate up- and down-regulated transcripts, respectively, from both control and cold treated tissues.

up-regulation suggests, *Gorai.002G251500* may play a significant role in activating downstream genes involved in the cold stress response, potentially through binding to specific cis-acting elements in the promoter regions of target genes.

## DEGs involved in signal transduction

Signal transduction pathways are essential for plants to perceive and respond to various environmental stimuli, including cold stress (*Yakel et al., 1995*; *Shi et al., 2012*). In this study, 35 out of the 136 key DEGs were associated with signal transduction processes.

The majority of these DEGs were annotated within the "plant hormone signal transduction (ko04075)" pathway. Specifically, 11 DEGs were associated with the jasmonic acid-mediated signaling pathway (GO:0009867), nine with the abscisic acid-activated signaling pathway (GO:0009738), and six were involved in the mitogen-activated protein kinase (MAPK) cascade (GO:0000165).

During the cold stress response, these signal transduction pathways work in a coordinated manner. For example, jasmonic acid-mediated signaling can modulate the expression of genes related to plant defense and stress adaptation. Abscisic acid-activated signaling plays a crucial role in regulating plant growth, development, and stress responses, especially in response to water-deficit and cold stress. The MAPK cascade is a highly conserved signaling pathway that can transmit extracellular signals to the nucleus, leading to the activation of stress responsive genes.

Of the 35 signal transduction related DEGs, 25 were up-regulated throughout the cold stress treatment in both *G. thurberi* and *G. trilobum*. One notable gene, *Gorai.008G230100*, which has a high expression level, is involved in "Starch and sucrose metabolism (ko00500)" and "maltose biosynthetic process (GO:0000024)". Based on annotations from the Swiss-Prot, GO, and KEGG databases, *Gorai.008G230100* is identified as Beta-amylase 3, a key gene in the plant's response to cold stress (GO:0009409). In contrast, nine DEGs were down-regulated across the cold stress treatment in both species, typically showing expression levels (Fig. 5B). The differential regulation of these genes in signal transduction pathways reflects the complexity of the cold stress response mechanism in cotton plants.

## DEGs involved in ROS production and scavenging

In plants, maintaining a balance between reactive oxygen species (ROS) production and scavenging is crucial for normal cellular function. However, under abiotic stress conditions such as cold stress, this balance is often disrupted, leading to oxidative stress (*Zhang et al., 2016*).

In this study, 16 DEGs were mapped to the "response to reactive oxygen species (GO:0000302)" term. When exposed to cold stress, some of these genes altered their expression levels, initially disturbing the ROS homeostasis. This disturbance was followed by compensatory changes in the expression of other genes aimed at restoring redox balance.

For example, an increase in the expression of genes involved in ROS production, such as certain oxidases, may lead to a transient accumulation of ROS. In response, genes encoding antioxidant enzymes, like SOD, POD, and CAT, are typically induced to scavenge the

excess ROS. This dynamic regulation of gene expression is a key mechanism that enables plants to mitigate oxidative damage under cold stress. The identification of these ROS-responsive DEGs provides valuable insights into the molecular mechanisms underlying redox homeostasis and cold stress tolerance in cotton.

## DEGs involved in carbohydrate transport and metabolism

Carbohydrate transport and metabolism play a central role in plants' response to cold stress (*Baba, Vishwakarma & Ashraf, 2017*). In this study, 12 out of the 136 DEGs were associated with carbohydrate related processes.

Among these, seven DEGs were down-regulated during the cold stress in both *G. thurberi* and *G. trilobum*. For example, *Gorai.001G239200*, which is involved in "Starch and sucrose metabolism (ko00500)" and "starch biosynthetic process (GO:0019252)", was down-regulated. This down-regulation led to a reduction in starch content, potentially as a strategy to increase the availability of soluble sugars for osmotic adjustment and energy production during cold stress.

Conversely, four DEGs were up-regulated in both species. Notably, *Gorai.008G230100*, also involved in "Starch and sucrose metabolism (ko00500)", was up-regulated, resulting in an enhanced soluble sugar content. Another gene, *Gorai.009G450300*, which is involved in "plant-type cell wall modification (GO:0009827)", showed different expression patterns between the two species; it was up-regulated in *G. trilobum* but down-regulated in *G. thurberi*. This differential regulation suggests a potential role for this gene in contributing to the varying degrees of cold resistance observed between *G. thurberi* and *G. trilobum* (Fig. 5D). Overall, these findings highlight the importance of carbohydrate metabolism in the cold stress response of cotton plants, with distinct DEGs regulating carbohydrate-related processes to adapt to cold stress.

## Hormones involved in the cold-stress adaptation

Plant hormones such as auxin (IAA), gibberellin (GA), abscisic acid (ABA) and ethylene (ETH) play crucial roles in regulating plant response to abiotic stress. To investigate hormone homeostasis under cold stress, we detect the content of four important hormones known to be involved in stress signaling pathways. Our analysis revealed a strong positive correlation between IAA and GA under cold stress ($R^2 > 0.93$, $P$-value $< 0.0034$), suggesting a significant crosstalk between these two hormones in modulating plant response to cold stress (Fig. 6). Additionally, GA levels were significantly correlated with PRO, POD and SS, highlighting the predominant role of GA signaling in cold stress tolerance in cotton. We further investigated interspecific differences in hormones levels between the two sister species. IAA content almost showed minimal difference between the two species, except at the 0 h time point. In contrast, GA and ETH showed significant differences between two species across all sampled points, while ABA levels were significantly different at 0, 12, 24, 48 h. These findings suggest that species-specific divergence in hormone regulation may underlie the differences in cold stress tolerance between the species. The variation in hormone levels driven by evolutionary divergence likely contributes to differential cold stress adaptation in these cotton species (Fig. 7).

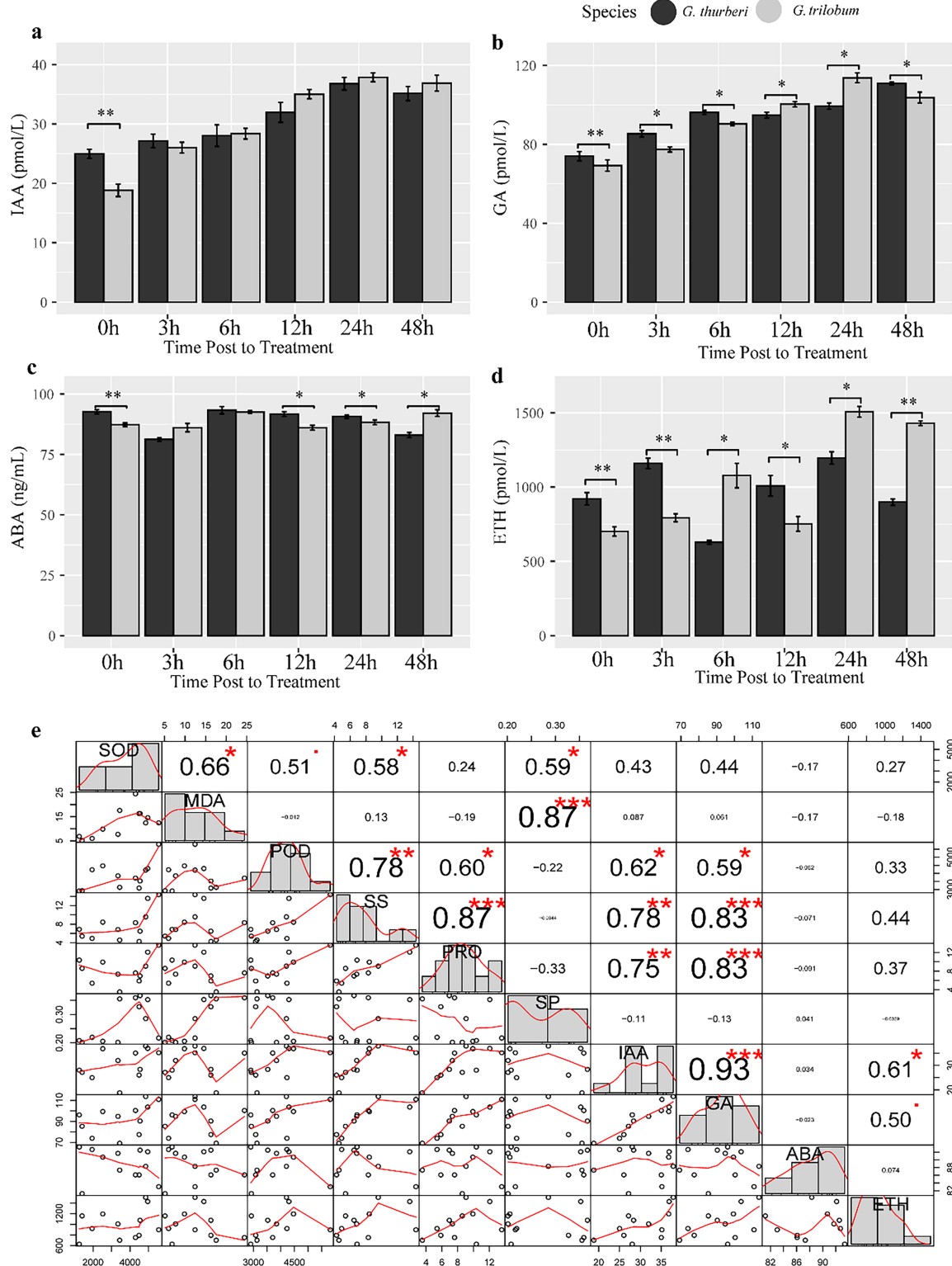

**Figure 6 Hormone assay and analysis under cold stress.** (A–D) The content of IAA, GA, ABA and ETH, respectively. (E) Correlation analysis of hormones. Multiple comparisons were performed with significant difference in different letter at $P < 0.05$ level; $*p < 0.05$ (significant), $**p < 0.01$ (highly significant), $***p < 0.001$ (extremely significant). Error bars represent SD.

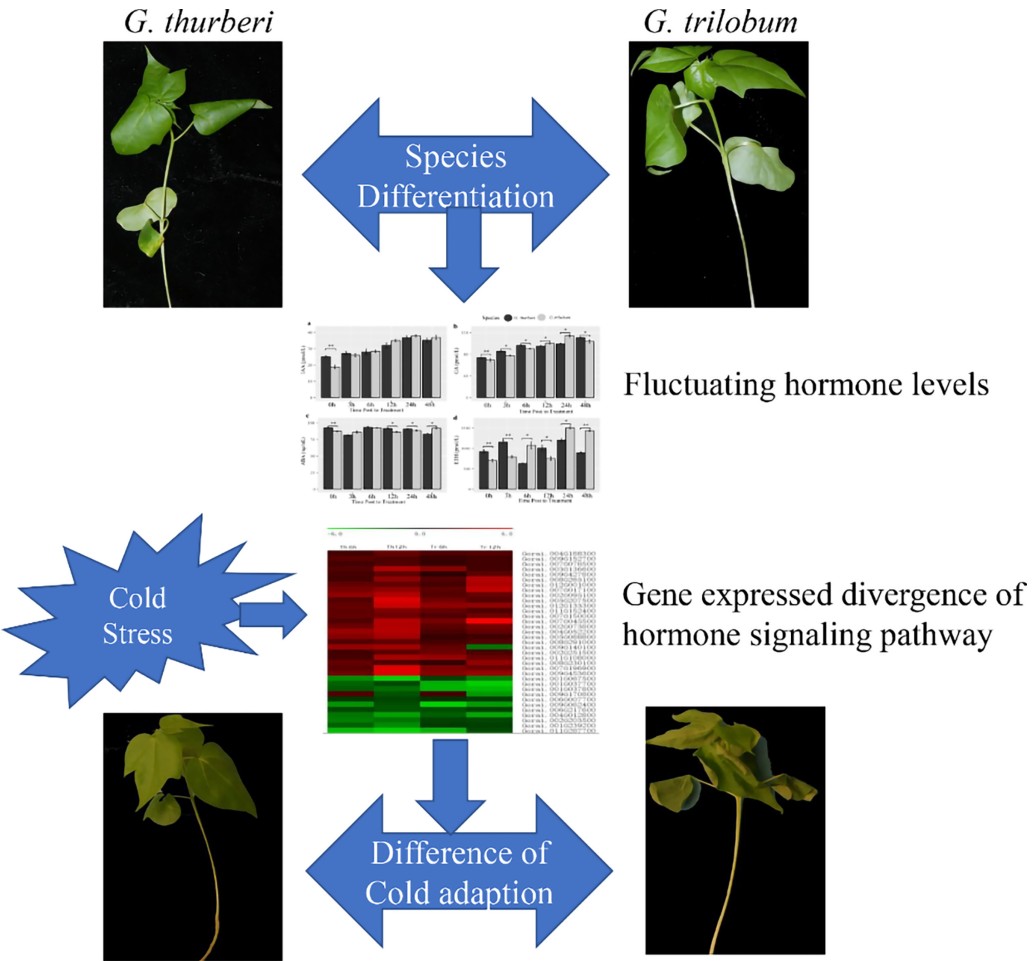

**Figure 7 Cold-adaptation difference of *G. thurberi* and *G. trilobum*.**

## Validation of the RNA-seq data by real-time qRT-PCR

To confirm the reliability of the RNA-Seq data, real-time quantitative reverse transcription polymerase chain reaction (qRT-PCR) analysis was conducted. Twenty genes were randomly selected from the RNA-Seq data, including six up-regulated genes, seven with unchanged expression, and seven down-regulated genes. Specific primers for these genes were designed (Table S4). The RNA used for qRT-PCR was extracted from the same samples as those used for RNA-Seq, ensuring consistency in the experimental materials.

The fold change values expressed as $Log_2(FC)$ obtained from both RNA-Seq and qRT-PCR were utilized to assess data reliability. Through correlation analysis, a strong positive correlation was observed between the qRT-PCR results and the RNA-Seq data. Under cold stress, at the 6 and 12-h time points in *G. thurberi*, the coefficients of determination ($R^2$) were 0.881 and 0.956, respectively. In *G. trilobum*, the $R^2$ values were 0.972 and 0.957 at the 6 and 12-h time points, respectively.

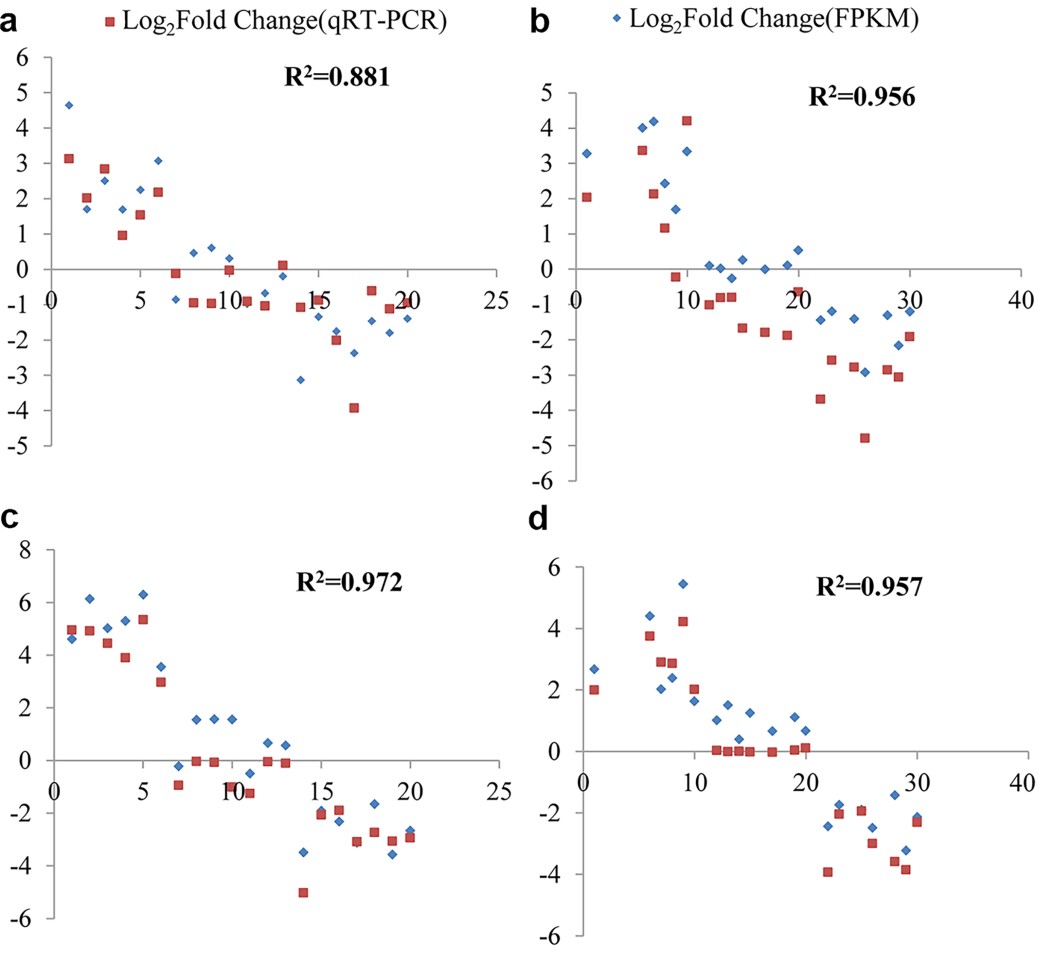

**Figure 8  qRT-PCR validation transcript levels evaluated by RNA-seq in *G. thurberi* and *G. trilobum* at 6 and 12 h under cold stress.** (A) The correlation of the qRT-PCR fold change and FPKM fold change at 6 h in *G. thurberi*. (B) The correlation of the qRT-PCR fold change and FPKM fold change at 12 h in *G. thurberi*. (C) The correlation of the qRT-PCR fold change and FPKM fold change at 6 h in *G. trilobum*. (D) The correlation of the qRT-PCR fold change and FPKM fold change at 12 h in *G. trilobum*. A total of 20 genes have been chosen: six up-regulated genes, seven unchanged genes and seven down-regulated genes.

Each sample was analyzed with three biological replicates in the real-time qRT-PCR experiment. Analysis of variance further demonstrated that there was no significant difference in the fold change values between qRT-PCR and RNA-Seq (Fig. 8). These results indicate a high degree of consistency between the gene expression profiles obtained from RNA-Seq and qRT-PCR, validating the reliability of the RNA-Seq data. This validation is crucial as it provides confidence in the gene expression changes identified through RNA-Seq, strengthening the overall findings and conclusions of this study on the cold-stress responses of *G. thurberi* and *G. trilobum*.

## DISCUSSION

Cold stress is a major abiotic factor that restricts cotton growth, development, and distribution (*Jung et al., 2017*). Understanding the molecular mechanisms underlying plants' responses to cold stress is crucial for improving crop tolerance. Although extensive research has been conducted on cold-adapted species such as *Arabidopsis thaliana*, rice, wheat, maize, and soybean (*Kim et al., 2001*; *Lee et al., 2001*; *Liu & Zhou, 2018*; *Robison, Yamasaki & Randall, 2019*; *Zhang et al., 2017*), cotton, a thermophilic crop sensitive to low temperatures, has received relatively less attention in this regard. Low temperatures, especially during the germination and seedling stages of cotton, can slow down growth, delay budding, flowering, and maturity, and ultimately impact yield and quality (*Butt et al., 2017*; *Zhu et al., 2013*). Moreover, previous studies on cotton's cold tolerance mechanism often focused on single samples, overlooking the shared response mechanisms among similar subspecies.

In this study, we compared the cold stress tolerance of two diploid wild cotton species, *G. thurberi* and *G. trilobum*. *G. thurberi* exhibited greater cold tolerance, as evidenced by its stable activities of SOD, POD, and higher PRO content under cold stress (*Elefteriou, 2018*). The content of soluble sugar increased gradually in both species after cold stress, indicating its role as a major protective molecule (*Liu et al., 2019*).

To explore the genetic basis of these differences, we performed transcriptome sequencing on leaf samples collected at 6 and 12 h after cold stress. The resulting reads were aligned to the *G. raimondii* reference genome for identification and annotation. In total, 23,726 and 23,820 genes were obtained for *G. thurberi* and *G. trilobum*, respectively. Additionally, 482 and 470 novel genes were discovered in *G. thurberi* and *G. trilobum*, and 445 genes were annotated by mapping to other species using GO, KEGG, KOG, and NR databases. Under cold stress and normal conditions, 4,227 and 4,174 differentially expressed genes (DEGs) were identified in *G. thurberi* and *G. trilobum*, respectively.

GO enrichment analysis showed that these DEGs were significantly enriched in single-organism processes, metabolic processes, response to stimuli, membrane-related components, organelle parts, catalytic activity, and nucleic acid-binding transcription factor terms in both species. KOG enrichment analysis further annotated the DEGs in terms of post-translational modification, protein turnover, chaperones, signal transduction mechanisms, and secondary metabolite biosynthesis, transport, and catabolism. Comparative analysis of DEGs between *G. thurberi* and *G. trilobum* identified 697 and 311 species-specific DEGs, respectively, along with 136 common DEGs at 6 and 12 h after cold stress. The enrichment analysis of these common DEGs was consistent with previous results, indicating that the GO terms, orthologous protein clusters, and cellular components play important roles in the cold stress response.

The plant response to cold stress involves a complex regulatory network (*Zhu, 2016*). Cold stress signals are first received by receptors on the plant cell membrane, followed by signal transduction through secondary messengers (*Shi, Ding & Yang, 2015*). This process

ultimately regulates the expression of genes involved in the cold stress response (*Liu & Zhou, 2018*). The network includes biological macromolecules such as enzymes and transcription factors (*Wang et al., 2017*), cellular components like the cell membrane and endoplasmic reticulum, and biological pathways related to stimulus response and signal transduction (*Kim et al., 2003*). It can regulate sugar transport and metabolism, protein synthesis, and active oxygen balance through $Ca^{2+}$, ROS, or hormone-mediated signal-transmitting pathways (*Monroy & Dhindsa, 1995*).

Further, our analysis suggests that GA may play a more dominant role than other hormones such as ABA, IAA, and ETH in cold stress response of *G. thurberi*. This could be due to GA's dual function in growth regulation and stress adaptation. Under cold stress, GA often balances growth suppression (to conserve energy) with stress signaling to maintain developmental plasticity. The up-regulation of GA-related biosynthesis or signaling genes may reflect a species-specific adaptation where *G. thurberi* fine-tunes its growth and metabolic processes more effectively than *G. trilobum*. Moreover, recent studies have shown that GA can interact antagonistically with ABA, influencing gene expression downstream of CBFs (*Achard et al., 2008*). In our findings, the differential regulation of GA-related genes may enhance cold resilience by modulating the hormonal crosstalk and improving energy allocation under stress. These aspects warrant further experimental validation.

Based on the 136 key DEGs identified in this study, we investigated the cold-stress-responsive signal-transmitting pathways. The CBF-regulated network, which has been well-studied in other crops such as Arabidopsis, rice, maize, and soybean. Notably, the CBF-regulated signaling pathway, a well-established mechanism in Arabidopsis for cold tolerance (*Zhu, 2016*; *Shi, Ding & Yang, 2015*; *Gu et al., 2016*; *Solanke & Sharma, 2008*), was also detected in cotton. While ICE1, a key upstream regulator in Arabidopsis, was not identified in our cotton transcriptome data, we detected ICE2 transcription factors, which may serve a similar regulatory function. The differential expression patterns of ICE2 between *G. thurberi* and *G. trilobum* suggest species-specific regulation of the CBF pathway. Similar to ICE1, ICE2 is a constitutively expressed protein, suggesting that it may regulate CBF gene expression through protein modifications. The different expression patterns of ICE2 in the two species indicate their different cold-stress-tolerance mechanisms.

We also identified key genes involved in ABA synthesis, NCED (9-cis-epoxycarotenoid dioxygenase) and AO (abscisic aldehyde oxidase), whose expression levels were up-regulated. ABA can participate in signal transduction by activating MAPK and regulating cold-stress-responsive genes. The number of MAPK-related DEGs in *G. trilobum* was less than that in *G. thurberi*, suggesting that cotton and Arabidopsis may have different cold stress response pathways, although the cold stress response network in cotton is similar to the drought stress response network in Arabidopsis. Additionally, our previous research (*Cai et al., 2019*) and the current study support the important role of the *CBF4* transcription factor in the cold stress response of diploid wild cotton. The up-regulation of *RAP2.7* (*Gorai.002G251500*), an AP2/ERF family transcription factor, in both *G. thurberi* and *G. trilobum* during cold stress further indicates its involvement in the CBF-regulated

network. These genes can be targeted in marker-assisted selection or introduced *via* genetic engineering into cultivated cotton to enhance its early season cold tolerance. Moreover, the differential expression of hormone related genes (ABA biosynthesis and signaling) and cold responsive regulatory elements provide candidate markers for functional validation in breeding programs. These insights pave the way for introgression of wild alleles into elite cultivars to improve germination, seedling vigor, and ultimately, yield stability under cold-prone environments.

## CONCLUSIONS

In this study, we utilized transcriptomic analysis to explore the gene expression patterns of two diploid cotton species under cold stress. A total of 4,227 and 4,174 DEGs were identified in *G. thurberi* and *G. trilobum*, respectively. Among them, 136 common DEGs were enriched in the Gene Ontology term "Carbon metabolism process". Plant hormones, particularly gibberellic acid (GA), were found to play a crucial role in modulating the plant's response to cold stress. GA showed a significant correlation with indole-3-acetic acid, proline, peroxidase, and soluble sugar, indicating that the GA signaling pathway is prominent in the cold-stress adaptation of *G. thurberi* and *G. trilobum*. *G. thurberi* demonstrated better cold tolerance, suggesting that the expression divergence between these two sister species contributes to the adaptive evolution of cotton in response to cold stress. These findings offer valuable insights into the molecular mechanisms underlying cold stress tolerance and provide a theoretical foundation for improving abiotic stress resistance in cultivated cotton by leveraging the genetic resources of wild cotton species.

## ACKNOWLEDGEMENTS

The authors would like to express gratitude to NCBI for providing data storage and sharing services.

### Funding

This work was supported by the National Natural Science Foundation of China, The National Key R&D Program of China (2022YFD1200304-4), the "STI 2030—Major Projects" Biological Breeding of Stress tolerant and High Yield Cotton Varieties (2023ZD04040-5) and the Autonomous Region "Science and Technology Innovation Team" (2023TSYCTD0003). The funders had no role in study design, data collection and analysis, decision to publish, or preparation of the manuscript.

### Grant Disclosures

The following grant information was disclosed by the authors:
National Natural Science Foundation of China.
The National Key R&D Program of China: 2022YFD1200304-4.
Biological Breeding of Stress tolerant and High Yield Cotton Varieties: 2023ZD04040-5.
Autonomous Region "Science and Technology Innovation Team": 2023TSYCTD0003.

## Competing Interests

Dong Wang is employed by Xinjiang Jinfengyuan seed industry Co., LTD.

## Author Contributions

- Dong Wang conceived and designed the experiments, performed the experiments, analyzed the data, prepared figures and/or tables, authored or reviewed drafts of the article, and approved the final draft.
- Juyun Zheng conceived and designed the experiments, prepared figures and/or tables, authored or reviewed drafts of the article, and approved the final draft.
- Ke Liu analyzed the data, authored or reviewed drafts of the article, and approved the final draft.
- Yanchao Xu performed the experiments, prepared figures and/or tables, authored or reviewed drafts of the article, and approved the final draft.
- Dingsha Jin conceived and designed the experiments, analyzed the data, prepared figures and/or tables, authored or reviewed drafts of the article, and approved the final draft.

## Data Availability

The data is available at NCBI: PRJNA554555.

## Supplemental Information

Supplemental information for this article can be found online at http://dx.doi.org/10.7717/peerj.19721#supplemental-information.

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
