# Peer review of "Gibberellic acid-mediated transcriptional divergence underlies cold stress adaptation in two diploid cotton species"

_PeerJ, doi:10.7717/peerj.19721_

## Round 0.1 · original submission · Major Revisions

Please address the concerns of all reviewers and revise the manuscript accordingly.

Reviewer 1 ·

Basic reporting

1. Clear and unambiguous, professional English used throughout.
Yes, the manuscript is well-written in formal academic English with only minor typographical errors (e.g., inconsistent gene formatting).

2. Literature references, sufficient field background/context provided.
Generally adequate, but could better integrate recent reviews on hormonal cold responses (e.g., Zhu 2016) and clarify gaps in cotton-specific mechanisms.

3. Professional article structure, figures, and tables. Raw data shared.
Structure and visuals are professional, but some figures lack error bars or statistical details (e.g., Figure 1). Raw data accessibility is unclear—no NCBI accession numbers provided.

4. Self-contained with relevant results to hypotheses.
Mostly yes, but claims about GA "driving" adaptation (Conclusions) overreach correlative data. Tighten links to original hypotheses.

Overall: Meets standards but requires minor refinements for full compliance.

Experimental design

1. Original primary research within the Aims and Scope of the journal:
Yes, this study represents original research on plant stress adaptation using comparative transcriptomics, which fits well within the scope of journals focused on plant biology, genomics, and environmental adaptation. The work investigates previously uncharacterized mechanisms of cold tolerance in wild cotton species.

2. Research question well defined, relevant & meaningful:
The research question is clearly stated: to understand the genetic basis of differential cold stress tolerance between G. thurberi and G. trilobum. This is highly relevant for cotton improvement programs. The introduction effectively identifies the knowledge gap regarding hormonal regulation (particularly GA) in cotton cold adaptation.

3. Rigorous investigation performed to a high technical & ethical standard:
The study employs:

Comprehensive physiological measurements

Replicated RNA-seq experiments (n=2 biological replicates)

Hormonal profiling

qRT-PCR validation
The methods appear technically sound, though additional replicates for RNA-seq would strengthen the findings. No ethical concerns are apparent.

4. Methods described with sufficient detail to replicate:
The methods section provides:

Detailed cold stress protocols

RNA extraction and library prep procedures

Bioinformatics pipelines (TopHat2, Cufflinks)

Hormone measurement techniques
However, some details could be improved:

Exact growth conditions (light intensity, humidity)

RNA-seq quality metrics (RIN values)

Full parameters for differential expression analysis

ELISA kit details (catalog numbers)

Suggested improvements:

Add more technical details about sequencing runs (e.g., read length, paired/single-end)

Include RNA quality metrics (RIN values) for all samples.

Specify all software versions and parameters.

Clarify statistical methods for physiological data analysis.

Provide catalog numbers for all commercial kits used.

The study generally meets these criteria well, but would benefit from additional methodological details to ensure full reproducibility.

Validity of the findings

1. Impact and Novelty:
The study demonstrates novelty through:

First comparative transcriptomic analysis of cold adaptation in these sister cotton species (Lines 58-60)

Identification of 445 newly annotated genes not present in the reference genome (Page 9, Figure 2)

Demonstration of GA-IAA crosstalk in cotton cold response (Figure 6, R²>0.93)

However, the impact could be strengthened by:

Explicit comparison to known cold response mechanisms in other crops

Discussion of how these findings could be translated to cultivated cotton

More detailed speculation about why GA plays a dominant role over other hormones

2. Data Robustness and Controls:
The study shows good experimental rigor:

Biological replicates for RNA-seq (n=2) and qPCR (n=3)

Multiple validation methods (physiological assays, qPCR, hormone measurements)

Use of FDR correction (≤0.01) for DEG identification

High correlation between RNA-seq and qPCR (R²=0.881-0.972)

Areas for improvement:

Statistical power calculation should be shown for RNA-seq design.

More replicates (n≥3) would strengthen RNA-seq conclusions.

Positive/negative controls for hormone assays should be described.

3. Conclusions:
The conclusions are well-supported by:

Direct linkage to the 136 common DEGs between species

Correlation between GA levels and physiological responses

Validation of key transcriptional changes

However, some conclusions could be tempered:

The statement about "driving adaptive evolution" (Line 460) oversteps the evidence.

The discussion of ICE2 (Lines 435-439) needs more comparative evidence.

Potential alternative explanations for the findings should be acknowledged.

Specific Recommendations:

Add a dedicated "Impact" paragraph discussing:

How could these wild relatives be used in breeding

Whether the GA pathway is conserved in polyploid cotton

Comparison to cold responses in other Malvaceae species

Strengthen statistical reporting:

Include exact p-values rather than just thresholds.

Show variance estimates for all physiological measurements.

Report effect sizes for key comparisons

Refine conclusions to:

Focus on observed correlations rather than evolutionary claims.

Acknowledge limitations of transcriptomics-only approach.

Highlight testable hypotheses generated by the work

The study makes a valuable contribution but would benefit from clearer framing of its significance and more cautious interpretation of mechanisms. The core findings about GA-mediated responses are well-supported and interesting for both basic and applied research.

Additional comments

This study provides valuable transcriptomic insights into the cold stress adaptation mechanisms of two diploid cotton species, Gossypium thurberi and G. trilobum, highlighting the role of gibberellic acid (GA)-mediated pathways. The experimental design is robust, with comprehensive physiological and molecular analyses, though some methodological details could be clarified. The findings are novel and relevant for cotton breeding, but the discussion could better integrate comparative perspectives from other plant systems. Overall, the manuscript is well-structured but requires minor refinements in data interpretation and presentation.
1. The title should explicitly mention "hormonal regulation" (e.g., "Gibberellic acid-mediated transcriptional divergence underlies cold stress adaptation in two diploid cotton species").
2. The statement that G. thurberi survives –6°C (Page 6) needs a citation—is this from field observations or controlled experiments?
3. Clarify why GA was prioritized over other hormones (e.g., ABA, ethylene) in the hypothesis.
4. The divergence time (0.7 million years, Page 6) should be contextualized with mutation rate estimates for Gossypium.
5. RNA-Seq Depth: The ~30–40M clean reads/sample (Table 1) are adequate but borderline for differential expression. Justify adequacy with a power calculation (e.g., using RNASeqPower).
6. Hormone Assays: Specify the ELISA kit catalog numbers and whether extraction included internal standards (e.g., deuterated GA).
7. Cold Stress Duration: Why were 6h and 12h chosen? Include a pilot time-course experiment to justify these timepoints.
8. Reference Genome: The G. raimondii genome (D5) is divergent from D1/D8. Were SNP-adjusted alignments performed to avoid mapping bias?
9. Figure 1: SOD Activity: The 3h spike in G. trilobum (Fig 1a) contradicts the conclusion of G. thurberi’s superior tolerance. Discuss this transient response.
a. MDA Data: Label the y-axis with units (nmol/g FW?).
10. Novel Genes (Figure 2): The 50.68% homology to G. arboreum (A genome) is puzzling for D-genome species. Could these be contamination or horizontal transfer?
11. DEG Analysis: The 136 common DEGs (Page 10) should be cross-referenced with known cold-responsive genes in Arabidopsis (e.g., CBFs, CORs).
a. The "hormone signal transduction" enrichment (Page 10) lacks specificity—lists top-ranked genes (e.g., DELLAs, PIFs).
12. Hormone Correlations (Figure 6): The R²>0.93 for GA-IAA is biologically implausible without feedback regulation. Check for overfitting or technical artifacts.
13. The ETH difference (Fig. 6d) is significant but unexplored—link to known ethylene-cold crosstalk (e.g., EIN3 regulation).
14. Figure 3: Chromosomal DEG distribution should highlight cold-responsive hotspots (e.g., near telomeres).
15. Figure 5:
a. The heatmap lacks a scale for log2FC—clarify if values are relative to the control or between species.
b. Gorai.008G230100 (Beta-amylase): Link its upregulation to soluble sugar accumulation (Fig. 1d).
16. Table 2: The >99% annotation rate is unusually high—was a Gossypium-specific database used?
17. GA’s Role: Contrast the GA findings with studies showing GA suppression under cold (e.g., in cereals). Is this divergence cotton-specific?
18. ICE2 vs. ICE1: The lack of ICE1 orthologs (Page 15) conflicts with its conservation in plants. Probe genome assembly gaps.
19. ROS Scavenging: The 16 ROS-related DEGs (Page 12) should be tied to SOD/POD activity trends (Fig 1a–c).
20. Carbohydrate Metabolism: The starch-to-sugar shift (Page 13) needs enzymatic validation (e.g., amylase activity assays).
21. qRT-PCR Primers: Provide primer sequences (Table S4) and amplification efficiencies.
22. Housekeeping Genes: Were UBQ or ACTIN tested for stability under cold stress? Cite a reference.
23. Negative Controls: Include a non-stressed time-zero sample for hormone assays to baseline circadian fluctuations.
24. FDR Correction: The FDR ≤ 0.01 threshold is stringent—was power compromised? Report the false-negative rate.
25. Batch Effects: Were RNA-Seq replicates sequenced in the same lane to avoid technical bias?
26. Hormone Extraction: Specify if leaf age/position was controlled, as hormone gradients exist.
27. Wild vs. Cultivated Cotton: Discuss whether the GA-mediated mechanism is conserved in G. hirsutum.
28. Arabidopsis Comparison: The CBF regulon (Page 15) is central in Arabidopsis but appears minor here—why?
29. "Expression divergence drives adaptation": This implies selection—use "correlates with" unless QTL evidence exists.
30. Novel Genes: Without functional tests, these are merely "unannotated"—not novel.
31. Abbreviations: Define DEGs, FPKM, and MDA at first use.
32. Gene Nomenclature: Italicize gene symbols (e.g., Gorai.002G251500).
33. Figure Legends: State n-values for replicates (e.g., "n = 3 biological replicates").

·

Basic reporting

-

Experimental design

-

Validity of the findings

-

Additional comments

The authors present a comprehensive analysis of diploid species of Gossypium thurberi and Gossypium trilobum in cotton, elucidating their response to cold stress tolerance. They tried to verify through RNA-seq analysis, physiological and biochemical, and RT-qPCR analysis. The study identifies DEGS and GOs that have significant expression differences in cotton seedlings. It is a good resource for cold stress tolerance in plants. However, I noticed typos, grammatical errors, and extra spaces after paragraphs and compound statements. Overall, the study provides valuable insights for cotton cold stress tolerance. Still, some errors need to be addressed before final acceptance.

Specific comments/suggestions:
Introduction
Scientific names should be italic starting from the abstract, corrections needed throughout the manuscript, including figures and references.
The text format of the manuscript should be justified.

Materials and methods
Please mention the accession number for NCBI GEO.

Results
Do you identify and recommend a few genes as candidates from the study for cold stress tolerance?
Improve the heatmap figures. It helps for comparison among genes.

Reference
In the reference section, there is a lack of consistency. Journal names are full and abbreviated. Follow the guidelines of the journal and maintain consistency.

Generally, the English and the arrangement of the manuscript should be improved. There are still some spelling and grammar mistakes. The authors need to check the paper thoroughly and fix all the typos.

·

Basic reporting

1 - “Upland cotton (G. hirsutum L.) accounts for approximately 90% of global production (Zhang et al. 2015). But intensive selection for yield and fiber quality has narrowed its genetic diversity (Xu et al. 2019).”
The sentence above should be rewritten. The punctuation is incorrect (but...)

2 - The relationship between gibberellic acid and cold stress should be described in a few sentences in the introduction.

3 - “under the same long-day conditions.”
Germination may have been ensured under long-day conditions. However, were the seedlings also grown under long-day conditions? Because cotton is a short-day plant.

4 - In which leaves of plants were the physiological and biochemical indices determined? Because these parameters may vary according to the development of the leaf. In the Method section, the developmental stage at which the observations were taken should be stated.

5 - It would be good to describe the software with which statistical analyses are carried out and figures are created.

6 - The discussion is written like a thesis. The study findings and the literature were not mutually evaluated. That's why there are disconnects in that sense. This section needs to be rewritten.

7 - Not fluent in English. Either get help from an expert or use artificial intelligence applications that correct grammar and make reading easier.

Experimental design

-

Validity of the findings

-

---

## Round 0.2 · accepted · Accept

All concerns of the reviewers were addressed, and the revised manuscript is acceptable now.

·

Basic reporting

Journal name: PeerJ

Title: Gibberellic Acid-Mediated transcriptional divergence underlies cold stress adaptation in two diploid cotton species

Wang et al. addressed reviewer comments and suggestions, including revising the title.

I see the authors use different font type and size in the manuscript.

Please address the following minor points and grammatical errors to polish the document for publication.

Experimental design

The submission address the research question.

Validity of the findings

The Gibberellic Acid signaling pathway plays a key role in cold-stress adaptation in G. thurberi and G. trilobum. The finding of the research contributes to adaptive evolution in cotton under cold stress.

·

Basic reporting

The other reviewers' and my suggestions have been meticulously answered. The article has been improved in all aspects. Congratulations.

Experimental design

Missing parts have been added, and unclear parts have been corrected.

Validity of the findings

It has been rewritten in a way that is understandable for readers.

Additional comments

The other reviewers' and my suggestions have been meticulously answered. The article has been improved in all aspects. Congratulations.